# Post-Translational Modification of MRE11: Its Implication in DDR and Diseases

**DOI:** 10.3390/genes12081158

**Published:** 2021-07-28

**Authors:** Ruiqing Lu, Han Zhang, Yi-Nan Jiang, Zhao-Qi Wang, Litao Sun, Zhong-Wei Zhou

**Affiliations:** 1School of Medicine, Sun Yat-Sen University, Shenzhen 518107, China; lurq@m.scnu.edu.cn (R.L.); imjiangyn@outlook.com (Y.-N.J.); 2Institute of Medical Biology, Chinese Academy of Medical Sciences and Peking Union Medical College; Kunming 650118, China; jennifer_z@imbcams.com.cn; 3Leibniz Institute on Aging–Fritz Lipmann Institute (FLI), 07745 Jena, Germany; zhao-qi.wang@leibniz-fli.de; 4Faculty of Biological Sciences, Friedrich-Schiller-University of Jena, 07745 Jena, Germany; 5School of Public Health (Shenzhen), Sun Yat-Sen University, Shenzhen 518107, China

**Keywords:** MRE11, PTM, DDR, disease

## Abstract

Maintaining genomic stability is vital for cells as well as individual organisms. The meiotic recombination-related gene *MRE11* (meiotic recombination 11) is essential for preserving genomic stability through its important roles in the resection of broken DNA ends, DNA damage response (DDR), DNA double-strand breaks (DSBs) repair, and telomere maintenance. The post-translational modifications (PTMs), such as phosphorylation, ubiquitination, and methylation, regulate directly the function of MRE11 and endow MRE11 with capabilities to respond to cellular processes in promptly, precisely, and with more diversified manners. Here in this paper, we focus primarily on the PTMs of MRE11 and their roles in DNA response and repair, maintenance of genomic stability, as well as their association with diseases such as cancer.

## 1. Introduction

Genetic information is faithfully transmitted from one cell to their progenies through its carrier DNA. However, DNA is frequently under attack by extrinsic and intrinsic agents, including reactive oxygen species (ROS), radiation, toxic chemicals, metabolites of cellular processes, and biochemical process of DNA, thereby resulting in base damage, mismatches, DNA strand breaks, as well as DNA-DNA or DNA-protein crosslinking, all of which are threats to genome integrity [1,2,3,4,5,6]. DNA double-strand breaks (DSBs) is the most toxic form of damage and a major threat to genomic stability. The cell is equipped with several effective DNA-repair mechanisms, including homologous recombination (HR), canonical non-homologous end-joining (NHEJ), and alternative nonhomologous end-joining (alt-NHEJ), to ensure the repair of DSBs. HR is a complex and relatively slow process involving multi-steps, which repair DSBs mainly occurring in the S-G2 phases with high-fidelity [7]. NHEJ or alt-NHEJ repairs DSBs predominantly in the G1 phase, although they can act throughout any phase of the cell cycle, by directly sealing the broken ends and is thus an error-prone pathway [7]. Nonetheless, NHEJ is a predominant repair mechanism in mammalian cells compared with microorganisms [8,9].

Meiotic recombination 11 (MRE11), the key components of the MRE11-RAD50-NBS1 (MRN) complex which function mainly to activate ATM- or ATR-mediated DNA damage response (DDR) upon DNA damage [10], play important roles in processing broken DNA ends and mediating an essential step for repairing damaged DNA [10]. Given its essential role in the repair of DSBs, the germline missense mutations in genes encoding the MRN complex cause human genomic instability syndromes such as: Ataxia-Telangiectasia-like disorder (ATLD, mutations in *MRE11*), Nijmegen breakage syndrome-like disorder (NBSLD, mutations in *RAD50*), and Nijmegen breakage syndrome (NBS, mutations in *NBS1*), respectively [11]. In addition to immunodeficiency, genomic instability, and hypersensitivity to radiation, cancer predisposition is one of marker symptoms for these syndromes [12,13,14]. Additionally, plenty of somatic mutations (Table 1), including missense, frameshift and stop code gained, of genes encoded in the MRN complex have been linked to many types of cancer, such as breast, ovarian, colorectal, gastric, and prostate cancers [15]. All of these suggest that MRE11 and its copartners play vital roles in preventing genomic instability and tumorigenesis.

Post-translational modifications (PTMs) refer to the chemical modifications of protein after translation, such as phosphorylation, ubiquitination, methylation, glycosylation, acetylation, nitrosylation, and so on [16,17,18]. PTMs participate in regulating the activity, structure, and function of proteins via the covalent addition of chemical groups, thus allowing for precise regulation of cellular activities [19]. Notably, accumulating evidence indicates that MRE11 can be modified by various PTMs which confer or fine-tune the regulation function of MRE11 in many biological processes, including DNA damage recognition, DNA binding ability, nuclease activity, and signal transmission ability. In the following sections, we summarize the recent progress in the discovery of all PTMs on MRE11 and the contribution of these PTMs to the biological and molecular functions of MRE11, and provide new insights into the future research directions to the physiological and pathological processes of MRE11’s PTMs.

## 2. Biological Functions of MRE11

The *MRE11* gene was first identified in *Saccharomyces cerevisiae* (*S. cerevisiae*) in 1993 as a meiotic recombination-related gene [20]. *MRE11* is highly conserved during evolution and its homologous genes have been found in fungi and archaea, as well as prokaryote and eukaryote kingdoms [20,21,22]. MRE11 proteins are largely involved in the processes of meiosis, DSBs’ repair, stabilization of replication forks, telomere maintenance, and defense against virus invasion (Figure 1).

### 2.1. MRE11 in Meiosis

The correct orientation of spindles and accurate separation of chromosomes in sexual reproduction organisms rely on the homologous chromosome recombination during meiosis, in which the formation of DSBs is a critical prerequisite to begin with. At the beginning of DSB formation, two molecules of topoisomerase-like protein (Spo11) attack the phosphate skeleton and covalently bind to the 5′-end of the DNA to form a protein-DNA adduct. MRE11 proteins then cut the DNA on both sides of the DSB to produce a nick by utilizing its 3′ to 5′ endonuclease activity, followed by a bidirectional resection of DSB by MRE11 and exonuclease 1 (EXO1) at the nick to cut off Spo11. Lastly, the DSB with about 300-nucleotide (nt) single strand is produced, allowing for other recombinant enzymes to complete the HR process (Figure 1A) [23,24,25]. The missense mutation of conserved Proline residue in the nuclease domain of *S. cerevisiae mre11* (mre11-P162S) results in DSBs forming a blockage, defective meiotic recombination, and thus low gamete activity [20], suggesting that MRE11 plays a key role during the meiosis of yeast.

### 2.2. MRE11 and DNA Damage Response

MRE11, together with RAD50 and NBS1, forms the MRN complex and acts as a sensor, transducer, and effector during DDR (Figure 1B). A high-throughput single-molecule imaging analysis revealed that the human MRN complex searches for free DNA ends by one-dimensional diffusion and three-dimensional collision, whereas MRE11 is a response to DNA end recognition and resection [26]. After recognizing the DSBs, the MRN complex recruits ataxia telangiectasia mutated (ATM) to the damaged sites and assists in ATM activation [27]. In turn, ATM phosphorylates all three members of the MRN complex and hundreds of other substrates [27,28] to mediate downstream signaling events including cell-cycle checkpoint control, DNA repair, senescence, cell death, and transcription activation [29]. During this process, ATM-dependent phosphorylation of MRE11 plays a pivotal role in determining the extent of end resection and is important for cell-cycle control and cell survival [30,31,32]. Being an effector, MRE11 trims DNA ends to start homologous recombination (HR) and promotes the HR process by releasing Ku and other DNA conjugates [7,26], or by holding sister chromatids together (Figure 1D) [33]. MRE11 is also required for nonhomologous end-joining (NHEJ) [34,35], a topic that will also be discussed in the following sections. 

### 2.3. MRE11 in V(D)J Recombination and CSR

In the immune system, DSBs are produced during V(D)J recombination and class switch recombination (CSR), which make important contributions to antibody diversity, allowing immune cells to recognize almost all kinds of antigens [36,37]. The variable region of immunoglobulin derived from B lymphocytes is encoded by V (variable), D (diversity) and J (joining) genes, V(D)J. Recombination which started with the DSB is made by the recombination-activating gene (RAG1–RAG2) recombinase, which would be repaired by the alt-NHEJ pathway [36]. Initially, the MRN complex and CtIP (CtBP-Interacting Protein) were recruited by PARP1 to start the 5′–3′ resection and, thereafter, breaks are repaired by alt-NHEJ (Figure 1C) [35,36,38]. In response to the antigen, CSR subsequently induces further modifications of immunoglobulin genes in B cells after V(D)J recombination. MRE11-RAD50-NBS1 is first recruited by DSB, initiated by AID (activation-induced cytidine deaminase), and then in turn activates ATM-dependent DDR (Figure 1C) [36]. Both classic NHEJ and alt-NHEJ pathways are used for CSR [39].

### 2.4. Other Function of MRE11

During DNA replication, obstructions to replication fork progression, especially in fragile sites, can cause fork stalling and facilitate the production of DSBs which is linked to genomic instability. To prevent the formation of DSBs, the MRE11-RAD50-XRS2 (MRX) complex, the yeast homolog of MRN, is recruited by the replication protein A (RPA) to stabilize the replication forks (Figure 1D) [40,41]. After the collapse of replication forks, the MRX complex can also be employed to the DSB sites of the stalled fork, which not only engages to hold sister chromatids at breaks but also promotes the collapsed replication forks to restart during DNA replication [40,42]. 

To maintain genomic integrity, proper spindle assembly and subsequent spindle dynamics are critical for accurate chromosome segregation. The MRN complex and CtIP are required for metaphase chromosome alignment, since the loss of MRN adversely affects spindle assembly and triggers a metaphase delay [43]. Further study has shown that the MRN complex, together with its partner, named mitosis-specific MRN-associated protein (MMAP), plays a role in spindle turnover, suggesting an important role of the MRN in the mitotic signaling pathway [44].

Notably, the maintenance of the telomere structure length requires the MRE11 complex, which participates in the cellular response to telomere dysfunction (Figure 1E) [45]. In yeast, the MRX, together with other proteins, executes the recruitment of telomerase to the shortened telomeres and cooperates with Rad6-Bre1-H2Bub1 to promote telomere-end resection, which thereby positively regulates telomerase- and recombination-dependent telomere replication [46,47]. In addition to DSBs in telomere, MRN helps to repair DSBs raising from hairpin DNA [48,49], and to cooperate with CtIP to remove lethal topoisomerase II-DNA adducts (Figure 1F) [50,51].

In addition to its response to and repair of DSBs raised from the host cell genomes, the MRN complex can bind to a viral genome and activate the ATM-mediated pathway during viral infection (Figure 1G) [52]. Viral-infection-activated MRN-ATM signaling defends the cells against DNA viruses and therefore offers an elegant mechanism to selectively prevent viral replication without jeopardizing the replication and viability of host cells [52]. This finding highlights a critical adaptation of the MRE11 complex to discriminate host and viral genomes, thus eliciting distinct responses.

## 3. MRE11 and Human Diseases

In humans, the *MRE11* gene has 20 exons at chromosome 11 and encodes a protein with 708 amino acids (aa). The MRE11 protein consists the N-terminal nuclease core (NC) domain that contains NBS1 interacting region (NIR), the RAD50- binding domain (RBD) in middle of C-terminal of NC, and two DNA binding domains (DBDs), with one in the N-terminus and the other in the C-terminus of RBD. In RBD, there is a GAR motif (glycine- and arginine-rich regions) which is an important region for arginine methylation (Figure 2A). In the middle of the NC domain and the C-terminal of DBD, there are “capping domains” that are responsible for the conformational change of MRE11 upon DNA binding (Figure 2A). The NC domain is highly conserved among different species, whereas the C-terminal region is variable [53,54,55,56]. Structurally, human MRE11 forms a U-shaped pocket dimer which binds to different DSBs in either a symmetric or asymmetric manner. The dimerization of MRE11 promotes its binding to DNA and plays a pivotal role in DSBs repair [57].

The dysfunction of MRE11, especially defects in its nuclease activity, is associated with many pathologies in humans. Multiple missense variants of *MRE11* have been linked to human diseases including progressive myoclonic ataxia (PMA, *MRE11*^A47V^) [58], Nijmegen breakage syndrome-like disorder (NBSLD, *MRE11*^D113G^) [59], and A-TLD [60,61,62,63,64,65] (Figure 2A). Interestingly, one variant of *MRE11* (c.657C > T, p.Asn219=), albeit synonymously, disrupts RNA splicing and triggers nonsense-mediated mRNA degradation, thus leading to A-TLD-like symptoms characterized as mental retardation, eye movement apraxia, gait ataxia, etc [65]. In patients with Alzheimer’s disease (AD), the protein levels of MRE11 and other components of MRN complex are substantially reduced in the neurons of brain cortex, suggesting that the loss of the MRN complex may be associated with the pathogenesis of AD [66].

In addition, the dysregulation of MRE11 is strongly associated with cancer. Firstly, somatic mutations of MRE11 have been identified in different types of cancer (Table 2). On the other hand, the abnormal expression of MRE11 has been shown to be involved in tumorigenesis and progression. In breast cancer cells, overexpression of MRE11 leads to cell proliferation by stimulating the STAT3 signaling pathway, and enhances the migration and invasion of cancer cells via activation of MMP-2 and MMP-9 [67]. Elevated expression of MRE11 also correlates with gastric cancer, colon cancer, and prostate cancer, conferring poor prognosis [68,69,70]. However, *Mre11* deficiency prevents tumor development in the p53-/- B-cell lymphoma xenograft assay [71]. Furthermore, MRE11 is supposed as a potential target for cancer therapy, since inhibitors to MRE11 exhibited promising therapeutic effects in cancer cells, either as favorable sensitizers or antitumor agents [72,73]. 

## 4. PTMs of MRE11 and Their Biological and Pathological Function

### 4.1. Phosphorylation of MRE11

Environmental factors, such as the radiation or chemicals, result in various types of DNA damage, among which DSB is the most lethal to cells. Once formed, DSBs are repaired via a serial action of DDR in order to maintain genome stability in cells. PTM is one of the major features during the process of DDR. Modified proteins participate in the recognition of DSBs and the amplification and transmission of primary signals, thereby regulating multiple cellular activities. Notably, the reversible phosphorylation of proteins at serine, threonine or tyrosine residues is the most widely abundant or most well-studied PTMs in DDR. The Ser-Gln (SQ) and Thr-Gln (TQ) motifs (SQ/TQ) are the preferred substrate motifs of the ATM and ATR kinases during DNA damage signaling [28] and MRE11 contains eight conserved SQ/TQ motifs (Figure 3A) which are potential substrates of ATM or ATR. Based on the database of PhosphoSitePlus [75], MRE11 can be phosphorylated at 38 sites, among which 14 sites had been clarified biological functions (Figure 3A). 

In response to DSBs, MRE11 recruits and activates ATM, and ATM in turn phosphorylates MRE11. Ionizing radiation (IR) and neocarzinostatin (NCS) produce DSBs, which induce the phosphorylation of MRE11 in the ATM wild-type (WT) cells, but not in ATM-deficient cells, indicating that the phosphorylation of MRE11 is dependent on ATM [76,77,78]. Specifically, ATM, upon IR, phosphorylates MRE11 at S676 and S678, which is critical for DSB repair and cell survival [78]. However, the ATM gene is not essential for the phosphorylation of MRE11 upon DSBs. For an instance, arsenic-induced DSBs trigger phosphorylation of MRE11 in an NBS1 but not in an ATM-dependent manner [76]. Moreover, application of the ATM-specific inhibitor KU55933 has no effect on MRE11 phosphorylation at S676 and S678 in *Xenopus* cell-free egg extracts with DSBs-containing DNA [79]. Likewise, ultraviolet radiation (UV) or methyl methanesulfonate (MMS) (both induce SSBs) induces phosphorylation of MRE11 independent of ATM [76,80], likely via other kinases, such as ATR, upon DSBs produced by *HaeIII* digestion [79]. All these suggest that MRE11 can be phosphorylated upon DNA damage in ATM-dependent and -independent manners, which may rely on the type of DNA damage. 

#### 4.1.1. MRE11 Phosphorylation Alters the DNA-Binding Capacity of MRE11

Phosphorylation of MRE11 at SQ/TQ sites has been showed to induce dissociation of the MRN complex from chromatin due to a reduction of MRE11 affinity for DNA (Figure 4A(1)) [79]. Similar results were observed when S649/S688 of MRE11 were phosphorylated, which significantly inhibits the loading of the MRN complex to DNA breaks and reduces recruitment of RAD50 and NBS1, leading to both the premature DNA damage checkpoint termination and the inhibition of DNA repair [32]. In chaperone-mediated autophagy (CMA) deficient cells, aberrant hyperphosphorylation of MRE11 causes the MRN complex to release from the DSBs, therefore increasing the accumulation of DNA damage [81]. Additionally, phosphorylation of MRE11 can alter its binding to DNA via affecting the interaction with other proteins. For example, MRE11 and RAD50 together with C1QBP form an MRC (MRE11-RAD50-C1QBP) complex, which stabilizes MRE11-RAD50 while inhibiting MRE11 nuclease activity by preventing its binding to chromatin or DNA [82]. Following DNA damage, ATM phosphorylates MRE11 at S676/S678 to dissociate the MRC complex and release MRE11-RAD50, allowing the assembly of the MRN complex to initiate DNA repair [82]. These findings indicate that phosphorylation of MRE11 regulates its affinity for DNA through direct as well as indirect ways (Figure 4A(1)).

#### 4.1.2. Phosphorylation of MRE11 Regulates HR and NHEJ Repair Pathway Choice

It is widely accepted that HR and NHEJ are two main pathways for the repair of DSBs [83,84,85]. Of note, both pathways are regulated by MRE11 phosphorylation. In response to IR, HR repair is defective in cells harboring two MRE11 variants (*MRE11*^S676A^ and *MRE11*^S678A^), although single-strand annealing is not affected [78]. It is worth noting that the phosphorylation of Mre11 plays a role to limit the ATM-mediated phosphorylation of EXO1. The impairing of EXO1 phosphorylation can accelerate the exonuclease activity of EXO1 and lead to uncontrolled resection, which can block HR repair (Figure 4A (2)) [78]. In yeast, the MRX complex also promotes DSB repair by HR in a phosphorylation-dependent manner. Removal of the phosphorylation sites in MRE11 and XRS2 results in a marked preference for DSBs’ repair via the NHEJ pathway, indicating that MRE11 phosphorylation functions specifically to inhibit NHEJ (Figure 4A (3)) [86]. Additionally, phosphorylation events of MRE11 are differently regulated by Tel1/ATM in response to DNA damage, or independently by Cdc28/Cdk1 kinases during mitosis which affect the cell-cycle progression and, consequently, the choice of DNA damage repair pathways [86], highlighting a central role of the MRE11 complex in the usage of DNA repair pathways is regulated by multiple phosphorylation signals. 

#### 4.1.3. MRE11 Phosphorylation Affects Cell Cycle and Chromosomal Alignment

As mentioned above, MRE11 can be phosphorylated by cell-cycle regulation kinase Cdc28/Cdk1 during mitosis [86], suggesting that the phosphorylation of Mre11 may play a role in cycle regulation (Figure 4A (4)). Moreover, MRE11 can be also phosphorylated at S649 by the kinase CK2 [87] or by the mitotic kinase Plk1 (polo-like kinase 1) [32]. Of interest, during G2 phase DNA damage checkpoint recovery, MRE11 is sequentially modified by Plk1-dependent priming phosphorylation at S649, followed by CK2-mediated phosphorylation at S688, leading to the release of MRN from DNA and the inactivation of both ATM-CHK2 (checkpoint kinase 2) and ATR-Chk1 signaling pathways [32]. In addition, Plk1-mediated phosphorylation of MRE11 at S688 also promotes its binding to MMAP (also known as C2orf44) to form the MMAP-MRN complex. The assembled MMAP-MRN complex further enables Plk1 to activate the microtubule depolymerase KIF2A (Identification of Kinesin Family Member 2A) and therefore induces spindle turnover as well as chromosome segregation during mitosis (Figure 4A (5)) [43,44]. It is interesting to note that MRE11 and its partner proteins in the MRN complex are shared between spindle alignment/checkpoint-regulation machinery and the DDR pathway. One possible explanation is that cells integrate the signals of both ATM-dependent phosphorylation and Plk1-dependent phosphorylation together via the MRN complex, to ensure its mitotic progression based on the status of genome integrity. As such, MRN could act at different phases of spindle dynamics including spindle assembly [43] and its turnover [44]. 

#### 4.1.4. MRE11 Phosphorylation in Tumorigenesis

The malfunction or deficiency of DDR results in genome instability, increasing the risk of cancer. The DDR pathway is generally turned off early in tumor development and cancer cell division may proceed even in the presence of damaged DNA. Indeed, as shown in one study, prostate cancer cells, which harbor a high activity of ribosomal S6 kinase (RSK), a downstream factor of the mitogen-activated protein kinase kinase (MEK) and the extracellular-regulated protein kinase (ERK), show an impairment of ATM-dependent G2/M checkpoint signaling and a resistance to DSBs [88]. MRE11 is phosphorylated at S676 by RSK, which disrupts MRE11 binding to DNA thereby blocks, ATM activation, and the phosphorylation of ATM’s downstream targets, NBS1 and H2AX [88]. This effect leads to the accumulation of DSBs and fuels cancer progression (Figure 4B) [88]. As a result, beyond the impact of MRE11 phosphorylation on cell-cycle checkpoints and DNA repair, Rsk-mediated phosphorylation of MRE11 may serve as a promising target for prostate cancer therapy.

In addition, deregulated AKT kinase activity due to PTEN deficiency also contributes to tumorigenesis. Interestingly, an increased AKT activity upon PTEN loss drives p70S6 kinase-mediated phosphorylation and the degradation of MRE11, thus impairing the MRN complex and the DDR in colorectal carcinoma cells [89]. In contrast, elevated AKT activity in primary human fibroblasts results in the accumulation of DNA damage, but reinforces oncogene-induced senescence, thus facilitating tumor suppression [89]. These findings shed light on a molecular mechanism by which the deregulation of AKT/mTORC1/p70S6K signaling exerts a profound effect on genome stability via MRE11 phosphorylation.

In summary, the phosphorylation of MRE11 is involved in many cellular activities and processes through affecting its affinity to DNA (Figure 4). To date, over ten phosphorylation sites have been investigated; however, their impact on the structure–function of MRE11, and the regulatory network of these phospho-sites, need to be further addressed in the future.

### 4.2. Ubiquitination and Ubiquitin-Like modification of MRE11

Ubiquitination (UB) and ubiquitin-like modification (UBL) are considered as modifiers for protein degradation, localization, and conformational changes. UBs and UBLs allow low molecular-weight proteins to covalently bond to lysine residues or other residues of substrates in the form of single or multiple molecular chains. This reaction requires three types of enzymes: E1 activating enzyme, E2 conjugating enzyme, and E3 ligase. In mammals, there is one major E1 (two in humans), 40 E2s, and over 600 E3s [90,91]. UBLs include small-UBL modification (SUMOylation), ubiquitin-fold modification (UFMylation), neddylation, etc., [91,92].

Except for phosphorylation, UB is the most frequently occurring PTM on MRE11 [75]. In response to DNA damage, MRE11 proteins are ubiquitinated, an important step for its binding to ubiquilin 4 (UBQLN4), a proteasomal shuttle factor [85]. Functionally, UBQLN4 is phosphorylated in an ATM-dependent manner and recruited to the DNA damage sites, where it impacts on DSB repair pathway choice toward NHEJ by repressing HR. Specifically, UBQLN4 interacts with ubiquitinated MRE11 to facilitate its proteasomal degradation (Figure 5A), since UBQLN4 depletion results in the accumulation of ubiquitinated MRE11 proteins at the DNA damage sites, leading to increased MRE11-dependent initiation of end-resection and subsequent shifting of more DSBs toward HR. In contrast, overexpression of UBQLN4 protects cells from genotoxic stress via reducing ubiquitinated MRE11 [85]. This discovery provides evidence of a role for UBQLN4-MRE11 in the UBQLN4 deficiency syndrome (UBDS) which is characterized by clinical symptoms, such as intellectual impairment, growth retardation, microcephaly, facial dysmorphism, hearing loss, ataxia, and anemia [85], which are also frequently detected in genome instability syndromes. 

Interestingly, elevated *UBQLN4* expression correlates with a poor overall survival in multiple human tumors (e.g., neuroblastoma, melanoma, ovarian, breast, and lung cancer) compared with low *UBQLB4* expression. Further analysis provides a rationale for this finding by showing that *UBQLN4* overexpression protects cells from acute genotoxic stress through enhancing the NHEJ-mediated sealing of DSBs, as NHEJ-driven mutagenesis is likely to be selected in cancer cells due to its error-prone mechanism [85]. Moreover, UBQLN4 binds to MRE11 and promotes MRE11 degradation, leading to cisplatin-resistance in cancer cells, i.e., esophageal squamous cell carcinoma (ESCC) [93]. Similarly, MRE11 ubiquitination is also involved in bladder cancer. An E3 ligase, cIAP2, is upregulated by HDAC inhibition in bladder cancer cells; elevated cIAP2 increases MRE11 turnover by potentiating its ubiquitination, resulting in the decreased ability of MRE11 to repair DSBs in DDR [94].

Like UB, SUMOylation has similar conjugation pathways but the process is carried out by SUMO-specific enzymes [91,95]. In *S. cerevisiae*, numerous HR proteins including MRE11 are SUMOylated upon DNA damage, parallel to other checkpoint-signaling cascades, thus accelerating DSB repair. Instead of targeting individual proteins for degradation, SUMOylation often targets entire protein groups, where it stabilizes physical interactions among the proteins [96]. Mechanistically, the yeast MRE11 non-covalently recruits the conjugated SUMO moieties to facilitate both global SUMOylation and DSBs’ repair. Particularly, there are two SUMO-interacting motifs (SIM1 and SIM2) (Figure 2B) within MRE11 that preferentially interact with the poly-SUMO chain. MRE11^SIM1^ is indispensable for MRX assembly, whereas MRE11^SIM2^ non-covalently links the MRX complex with the SUMO enzyme complex to promote the global SUMOylation of proteins that are responding to HR DNA repair upon methyl methanesulfonate (MMS) treatments [97]. Given its role to protect cells against DNA damage, MRE11 becomes a key target attacked by viruses. During adenovirus (Ad) infection, the ends of the Ad linear dsDNA are recognized by the host as DNA breaks, eliciting a DDR to allow the formation of concatemers by ligating Ad genomes (Figure 5A) [98,99,100]. Since the DDR normally inhibits viral DNA replication, Ad has developed different mechanisms to protect itself by the inactivation of the DDR pathway. For example, Ad encodes proteins that could either inactivate the MRN complex by directing UB-mediated degradation [100,101,102] or modulate the SUMOylation of MRE11 and NBS1 to inhibit the DDR pathway [103,104] (Figure 5A). Consequently, viruses fully exploit the host UB and SUMO machineries to optimize their cellular environment by inhibiting the induction of DDR during infection. 

In addition to SUMOylation, UFMylation is one of the new additions to the UBLs. Similarly, UFMylation has conjugation pathways with a three-step enzymatic reaction and has been linked to several cellular activities [105,106,107]. Since MRE11 UFMylation has just recently been identified, thus data are very limited. Under physiological conditions, the UFMylation of MRE11 promotes the MRN complex formation and ensures timely recruitment of the MRN complex to the DNA damage sites (Figure 5B). In response to DSBs, MRE11 is UFMylated on lysine 282 (K282), which is required for optimal ATM activation, HR-mediated repair and genomic stability [31] (Figure 5B). Interestingly, a pathogenic variant of MRE11 (*MRE11*^G285C^) in uterine endometrioid carcinoma displays a similar cellular phenotype with the UFMylation-defective variant of MRE11 (*MRE11*^K282R^), implying that MRE11 UFMylation is very likely to be associated with tumorigenesis [31]. A new study shows that the UFMylation of MRE11 on K281 and K282 is responsible for MRE11 interaction with the telomere protein TRF2 and is essential in maintaning telomere length and aiding cell survival [108]. 

Despite a similar enzymatic reaction, UBs and UBLs lead to different cell fates. Whereas UB leads to the proteasomal degradation of MRE11, SUMOylation regulates the subcellular localization of MRE11 [103,104]. UFMylation neither degrades nor stabilizes MRE11 proteins but functions to control the formation of the MRN complex as well as to maintain telomere length. Nevertheless, compared with phosphorylation, further in-depth studies of MRE11 on its UBs and UBLs are urgently needed.

### 4.3. Methylation of MRE11

Protein methylation is regarded as a highly versatile, pervasive, and reversible way of protein modification. In eukaryotes, most protein methylation is carried out by two enzyme families, lysine methyltransferases (KMTs) and protein arginine methyltransferases (PRMTs), which transfer methyl marks to the ε amino group of lysine and the guanidinium group of arginine, respectively. In humans, methylation of both lysine and arginine residues are abundant, whereas methylation of other amino acid residues has been described but with very limited understanding [109]. 

PRMTs utilize S-5′-adenosyl-L-methionine (SAM) as a methyl donor to catalyze monomethylation and dimethylation at arginine residues. The latter is further divided into symmetric dimethylation and asymmetric dimethylation (Figure 6A) [110]. PRMT-mediated arginine methylation usually occurs at GAR (Glycine/arginine-rich) motifs, which are methylated by PRMT1, PRMT3, PRMT5, PRMT6, and PRMT8 [111,112]. Notably, MRE11 bears a GAR motif that is conserved among different eukaryotic species, suggesting a possibility of MRE11 arginine methylation by PRMTs (Figure 3C). Indeed, an early proteomic study first identified MRE11 as an arginine-methylated protein using arginine methyl-specific antibodies [113]. Subsequent studies revealed that MRE11 contains asymmetrical dimethyl arginine within its GAR motif that is necessary for MRE11 exonuclease activity. Specifically, MRE11 is arginine-methylated by PRMT1, and this methylation is required for intra-S-phase DNA damage checkpoint response [114]. Interestingly, mutations of the arginine within the GAR motif severely impair the 3′ to 5′ exonuclease activity of MRE11 without affecting its ability to form complexes with RAD50 and NBS1 (Figure 6B (1)) [114]. Further investigation provided evidence that PRMT1 interacts with MRE11 but not the MRN complex, indicating that MRE11 arginine methylation occurs prior to the association with RAD50 and NBS1. Importantly, the MRE11-methylated GAR motif is sufficient for its targeting to DNA damage foci and colocalization with γH2AX, which are necessary for sensing DSBs and initiating DDR (Figure 6B (2)) [115,116,117]. Nevertheless, the specific mechanism of how arginine methylation of the GAR motif regulates exonuclease activity of MRE11 remains unclear. One hypothesis is that the methylation of the GAR motif alters its protein conformation [118] which may thereby affect directly its enzyme activity or generate a docking site for certain proteins that can indirectly regulate MRE11 activity.

In addition to the MRE11-PRMT1 interaction, TIS21 and GFI1 were further identified as critical regulators of methylation. TIS21 accelerates DNA repair by enhancing PRMT1 activity and MRE11 methylation, leading to MRE11 activation in vitro and in vivo [119]. Likewise, GFI1 enables PRMT1 to bind and methylate MRE11 in T lymphocytes, thereby facilitating efficient DNA repair [120]. Moreover, the methylation of human MRE11 also contributes to the promotion of alt-NHEJ, one of the NHEJ sub-pathways, providing new insights into the mechanisms by which MRE11 regulates the accuracy of DSB repair via control of the alt-NHEJ sub-pathway (Figure 6B (3)) [121].

Mice, which harbor homozygous arginine (R) to lysine (K) mutations in the GAR motif of MRE11--namely *Mre11*^RK/RK^, showed no overt phenotypes. However, upon challenge with IR, the *Mre11*^RK/RK^ mice and their mouse embryonic fibroblasts (MEFs) display defective cell-cycle checkpoint and aberrant chromosomes, demonstrating a key role for the GAR motif in maintaining genomic stability (Figure 6B (4–5)). Mechanistically, the *Mre11*^RK/RK^ cells display an impairment of ATR/CHK1 signaling activation and recruitment of RPA and RAD51 proteins to the DSB sites in response to IR, whereas the MRN complex formation, localization of MRN to the DSBs sites, and the ATM activation are normal (Figure 6B (6)) [122]. It is possible that the chosen lysine maintains a positive charge of the residue, keeping its effects on these processes. 

In *Drosophila*, mre11 is also methylated by *Drosophila* arginine methyltransferase 1 (DART1), the fly homolog of PRMT1, at multiple arginine in the GAR motif. Strikingly, the mre11-DART1 interaction in *Drosophila* does not require the GAR motif in S2 cells [123]. Further in vivo studies showed that though both the *mre11*^RA^ (the single arginine>alanine variant) and *mre11*^4RA^ (all arginine variants) flies are sensitive to IR, the *mre11*^RA^ flies display no IR-induced G2/M cell-cycle checkpoint defect in wing disc and eye disc, which is different to that of murine *Mre11*^RK/RK^ who show the flies as both sensitive to IR and lacking G2/M checkpoint [122]. As such, the underlying physiological significance of arginine methylation of *Drosophila* mre11 in mice is various. 

In contrast to eukaryotes, the GAR motif is missing in the archaeal and bacterial Mre11 orthologs. Instead, the archaeal MR complex exhibits extensive lysine methylations and glutamate/aspartate methylations of both Mre11 and Rad50 proteins, which were identified under physiological conditions [124]. Among them, the methylation of one Mre11 aspartate residue 84(D84) is predicted to reduce its nuclease activity, suggesting archaeal MRE11 methylation may play role in the regulation of the MR complex and DDR [124]. However, future studies are required to identify which methyltransferase(s)/demethylase(s) are involved in the regulation and the balance of lysine methylation of MRE11, and to determine the exact biological roles of these methylation sites. 

In fact, the metabolic cost of methylation is very high, where 12 ATP equivalents are required to transfer to a methyl group [118]. Therefore, a methylation event will very likely not survive during evolution if it is not essential to cells. The fact that MRE11 methylation occurs under physiological and stressed conditions across microorganisms, vertebrates and invertebrates, highlights the importance of this PTM in maintaining cellular genome stability (Figure 6B).

## 5. Concluding Remarks and Future Directions

Compared to the high consumption of energy and relatively long-term feedback mode of the DNA-RNA-protein synthesizing system, PTMs, by altering structure and function of protein, make a timely response and confer diversity as well as specificity on regulating protein function during different biological processes. Given the overwhelming number of scientific explorations [125,126,127,128,129], defects in PTMs have been linked to numerous developmental disorders and regarded as an indispensable factor in pathogenesis of human diseases.

Being a defensive player, MRE11 is one of the substrates regulated by multiple PTMs following DSBs. However, our understanding of other MRE11 PTMs, apart from phosphorylation, is rather incomplete. All PTMs of MRE11 seem to play essential and diverse roles in selection of DNA repair pathways and signaling. For example, MRE11 phosphorylation specifically inhibits NHEJ [86], whereas the methylated and ubiquitinated MRE11 contribute to the promotion of alt-NHEJ [121] and NHEJ [85], respectively. The dynamic regulation of these two mechanisms by different PTMs on MRE11 offers a flexible choice of cell fates. Imbalance of NHEJ and HR pathways in mammalian cells may disturb the cellular processes and functionality of a cell, leading to genomic instability and pathological events, such as cancers. 

Despite increasing research having greatly expanded our knowledge underlying the biological functions of MRE11 and their PTMs, there remains key scientific issues that need to be addressed. For example, nearly 100 PTM sites on MRE11 have been identified, but most of them, such as acetylation [130,131], remain mysterious (Figure 3). In addition, some PTMs, such as UB, SUMOylation, and acetylation, have been found to occur on the same residue of MRE11 proteins (Figure 3B) [132,133,134]; however, it is unclear how MRE11 receives and integrates those signals, and what the crosstalk is among them. Despite the known structure of MRE11 [57], the function(s) of its key motifs, its conformational change caused by PTMs, as well as the relationship between disease-associated variants and its PTMs, remain unclarified. Furthermore, although it has been shown that some reversibility of Mre11 PTMs taken place, such as dephosphorylation, the biological meaning of the reversible process is still unclear. Therefore, the function of enzymes that respond to removing the PTMs of MRE11, as well as biological consequence of blockage of the reversible process, if there is one, are required for further investigation. Inspiringly, although several compounds targeting the MRE11 or MRN complexes, as well as their interactive partners, have been identified in preclinical studies with encouraging results [135], PTM sites of MRE11 could have novel potential for precision medical designing, since PTMs are catalyzed by different enzymes which can be targeted by either chemical compounds or genetic manipulation. Therefore, further effort is required to dissect the biological function of MRE11 PTMs and their regulation mechanism, which will promote a better understanding of the initiation and development of MRE11-associated human diseases and define a novel path to well-tailored precision therapies. 

## Figures and Tables

**Figure 1 genes-12-01158-f001:**
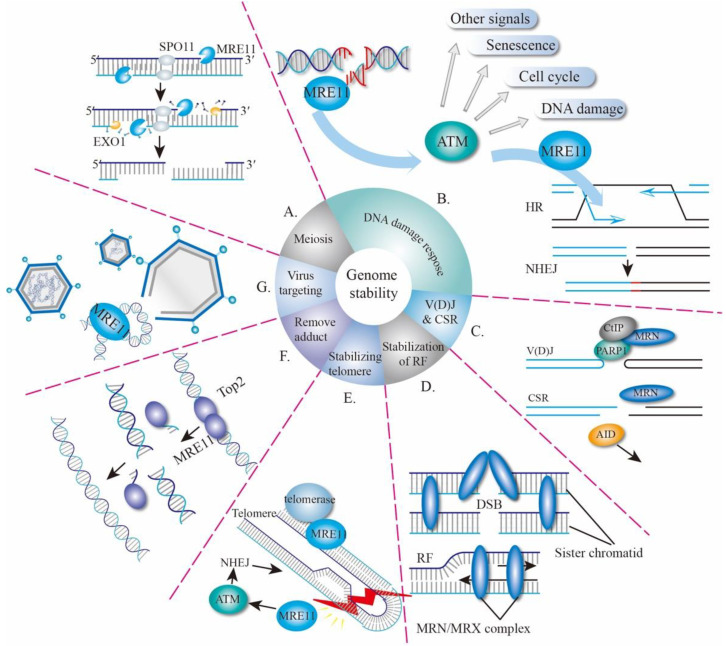
Biological Functions of MRE11. (**A**) MRE11 is involved in meiosis to remove Spo11 with endonuclease activity. (**B**) MRE11 is involved in whole DNA damage response, including DSB recognition, signaling and repair. (**C**) CSR and V(D)J recombination. (**D**) Stabilization of replication forks. (**E**) Stabilizing telomere. (**F**) Removal of toxic DNA adduct. (**G**) Virus targeting. CSR, class switch recombination; V(D)J, V (variable), D (diversity), J (joining) genes; RF, replication fork; HR, homologous repair; NHEJ, non-homologous end-joining; DSB, DNA double-stranded break.

**Figure 2 genes-12-01158-f002:**
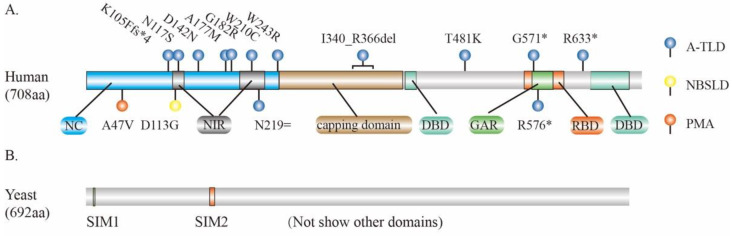
Key domains and mutations of MRE11. (**A**) Key domains and mutation sites of MRE11 for human genetic diseases. (**B**) Two SUMO-interacting motifs (SIM1 and SIM2) within MRE11 of *S. cerevisiae* that preferentially interact with the poly-SUMO chain. PMA, progressive myoclonic ataxia; NBSLD, Nijmegen breakage syndrome-like disease; A-TLD, ataxia-telangiectasia-like disorder; NC, nuclease core; DBD, DNA binding domains; GAR, glycine- and arginine-rich region; NIR, NBS1 interacting region; RBD, RAD50 binding domain; del, deletion; asterisk (*) means stop codon; equal sign (=) means synonymous mutation; fs*4 means stop codon is the 4th amino acid following frameshift mutation.

**Figure 3 genes-12-01158-f003:**
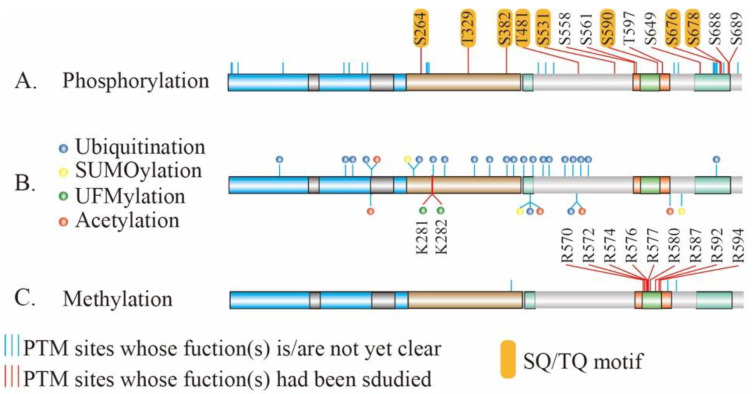
The PTMs sites on MRE11. The validated and potential PTM sites of MRE11. PTM sites with unclear function(s) are marked by blue line; PTM sites with validated functions are marked by the red line. The first bar shows phosphorylation sites of MRE11 (**A**); the middle one ubiquitination, SUMOylation, UFMylation, and acetylation of MRE11 are shown with blue-, yellow-, green-, and red-filled circles respectively (**B**); methylation PTM sites of MRE11 is shown in the third bar (**C**). SQ/TQ, Ser-Gln (SQ) and Thr-Gln (TQ).

**Figure 4 genes-12-01158-f004:**
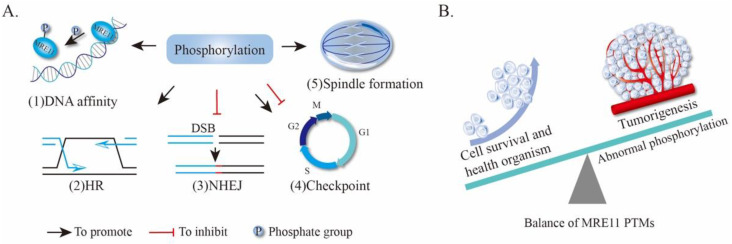
The effect of MRE11 phosphorylation. (**A**) The effect of MRE11 phosphorylation on DNA affinity, HR, NHEJ, cell cycle checkpoint, and spindle formation. (**B**) Abnormal phosphorylation induces tumorigenesis. HR, homologous repair; NHEJ, non-homologous end-joining; DSB, DNA double-stranded break.

**Figure 5 genes-12-01158-f005:**
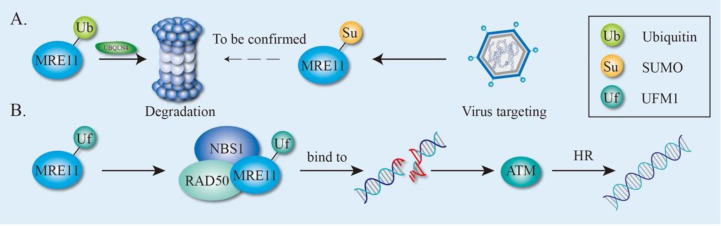
The effect of ubiquitination and ubiquitin-like modification of MRE11. (**A**) The effect of MRE11 ubiquitination and SUMOylation. MRE11 ubiquitination promotes its degradation mediated by UBQLN4 through proteasome and SUMO-MRE11 is targeted by adenovirus. (**B**) The effect of MRE11 UFMylation. MRE11 UFMylation promotes MRN complex formation and DNA repair. HR, homologous repair.

**Figure 6 genes-12-01158-f006:**
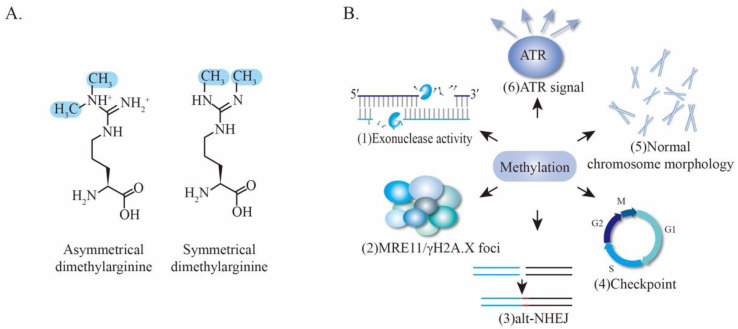
The effect of methylation of MRE11. (**A**) Asymmetric methylation and symmetric methylation on arginine. (**B**) The effect of MRE11 methylation, including regulating exonuclease activity, forming DNA repair foci, promotion of alt-NHEJ, regulating cell-cycle checkpoint, maintaining normal chromosome morphology, signaling of ATR, and so on. Note that the image of (3)alt-NHEJ shows a simplified progress, and is different from canonical NHEJ.

**Table 1 genes-12-01158-t001:** Summary of total cancer-associated mutations in each component of MRN complex.

Genes	Number of Somatic Mutations *	Affected Cases
*MRE11*	125	840
*RAD50*	229	869
*NBS1*	136	978

*: Including missense, frameshift and stop code gained mutation. Data were obtained by searching the TCGA database (https://portal.gdc.cancer.gov/) (accessed on 26 June 2021) with the indicated gene name. “Number of somatic mutations” and “Affected cases” were extracted from the “Most Frequent Somatic Mutations” section.

**Table 2 genes-12-01158-t002:** Association of MRE11 mutations with different types of cancer.

Cancer Type	Protein Change	Mutation Type	Variant Type
Acute Myeloid Leukemia	Q477E	Missense_Mutation	SNP
Adrenocortical Carcinoma	R351C	Missense_Mutation	SNP
Bladder Urothelial Carcinoma	Q438E	Missense_Mutation	SNP
Breast Invasive Ductal Carcinoma	R503C	Missense_Mutation	SNP
Cervical Squamous Cell Carcinoma	S641*	Nonsense_Mutation	SNP
Cutaneous Melanoma	N511Ifs*13	Frame_Shift_Del	DEL
Glioblastoma Multiforme	R364*	Nonsense_Mutation	SNP
Head and Neck Squamous Cell Carcinoma	S608*	Nonsense_Mutation	SNP
Lung Squamous Cell Carcinoma	E350*	Nonsense_Mutation	SNP
Mucinous Adenocarcinoma of the Colon and Rectum	E460*	Nonsense_Mutation	SNP
Pancreatic Adenocarcinoma	D498N	Missense_Mutation	SNP
Papillary Renal Cell Carcinoma	L44V	Missense_Mutation	SNP
Papillary Thyroid Cancer	R505I	Missense_Mutation	SNP
Renal Clear Cell Carcinoma	T627A	Missense_Mutation	SNP
Serous Ovarian Cancer	A526Gfs*16	Frame_Shift_Ins	INS
Stomach Adenocarcinoma	N511Ifs*13	Frame_Shift_Del	DEL
Uterine Endometrioid Carcinoma	E460*	Nonsense_Mutation	SNP

Del/DEL, deletion; asterisk (*) means stop codon; fs*13 means stop codon is the 13th amino acid following frameshift mutation; Ins, insert; SNP, single nucleotide polymorphism. Data were obtained from the database cBioPortal by searching the item “MRE11” in “Quick search” and then clicking “Mutations” [74]. Mutations identified in represent types of cancer were summarized.

## Data Availability

The data presented in this study are available on request from the corresponding authors.

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
