# Peer review of "Post-Translational Modification of MRE11: Its Implication in DDR and Diseases"

_genes, 2021, doi:10.3390/genes12081158_

Round 1

Reviewer 1 Report

In this review, Lu et al. summarized the role of post-translational modification of MRE11. The manuscript is well-written and covers most recent reports related to the function of MRE11. However, I have several minor comments to confirm the intended meanings in the manuscript. In addition, I would suggest that additional figures should be prepared to clarify the message in each section.

1. Line 32, “NHEJ is the only pathway for DSB repair …,” it is exaggerated to say that NHEJ is “the only pathway” for DSB repair in the G1 phase. The authors should address the possibility of other pathways, for example, alt-NHEJ, or simply rephrase, stating that NHEJ is the primary pathway in normal cells.

NHEJ is used in S/G2 as well as G1, i.e., throughout the cell cycle phase. This should be amended.

2. Line 51, “Data is obtained from TCGA database.” Is this new data? If the authors summarized this data for this manuscript, the method should be described.

3. Line 55 and 60, the authors used “etc”, but I am unsure of the intended meaning. The authors should add the rest of the possibilities.

4. Line 82, a reference (PMID: 28059759) can be added.

5. Line 134, “MRE11 and MRN play roles to repair…,” MRE11 is included in MRN complex. This should be spelled out.

6. In the section “Biological functions of MRE11” why didn’t the authors mention VDJ and CSR? The authors should add overviews of them.

7. Line 414, I am unsure that D-NHEJ is commonly used in this field. D-NHEJ generally means DNA-PK (DNA-dependent protein kinase)-dependent pathway of NHEJ; although this term is not widely used in our field. To avoid any confusion for readers, the authors should simply use “deleterious NHEJ” instead of D-NHEJ throughout the manuscript.

8. Line 417, I cannot find the origin of the abbreviation “RK/RK.”

9. Some editorial errors should be corrected, for example, [38][39] -> [38, 39] and tumor genesis -> tumorigenesis.

10. The text is well written throughout the manuscript; however unfortunately, the quality of the figures is poor. Each figure does not appropriately represent the summary of the text. If space is not limited, Figure 2 should be separated to clarify the role of each PTM against MRE11, i.e., structure (Figure 2), phosphorylation (Figure 3), ubiquitination (Figure 4), and methylation (Figure 5). For example, by looking at Figure 2 in the current version, it is unclear whether the phosphorylation of MRE11 activates its nuclease activity or the recruitment of MRN at DSB sites. And also, for example, it is unclear whether cell viability and tumorigenesis are downstream of HR and NHEJ. I am unsure why these are drawn in parallel arrows. In addition, if a PTM affects other PTM events, the interplay should be also discussed in the text or figures.

Author Response

Reply to Reviewer #1

Comments and Suggestions for Authors

In this review, Lu et al. summarized the role of post-translational modification of MRE11. The manuscript is well-written and covers most recent reports related to the function of MRE11. However, I have several minor comments to confirm the intended meanings in the manuscript. In addition, I would suggest that additional figures should be prepared to clarify the message in each section.

Responses:

We appreciated this reviewer for her/his positive view of our study and for his/her constructive comments on the manuscript.

  1. Line 32, “NHEJ is the only pathway for DSB repair …,” it is exaggerated to say that NHEJ is “the only pathway” for DSB repair in the G1 phase. The authors should address the possibility of other pathways, for example, alt-NHEJ, or simply rephrase, stating that NHEJ is the primary pathway in normal cells.

NHEJ is used in S/G2 as well as G1, i.e., throughout the cell cycle phase. This should be amended.

Responses:

We are in total agreement with this reviewer. We now correct our statement and rewrite them in the Line 30~Line37 of the revised manuscript as:
“The cell is equipped with several effective DNA repair mechanisms, including homologous recombination (HR), canonical non-homologous end-joining (NHEJ) and alternative nonhomologous end-joining (alt-NHEJ), to ensure the repair of DSBs. HR is a complex and relatively slow process involving multi-steps, which repairs DSBs mainly occurring in S-G2 phases with high-fidelity. NHEJ or alt-NHEJ repairs DSBs predominantly in G1 phase, although they can act throughout any phase of the cell cycle, by directly sealing the broken ends and therefore is error-prone pathway.”

  1. Line 51, “Data is obtained from TCGA database.” Is this new data? If the authors summarized this data for this manuscript, the method should be described.

Responses:

Indeed, this table is a summary of the number of “somatic mutations” and the reported “affected cases” with relative mutations. Data was obtained by searching the TCGA database(https://portal.gdc.cancer.gov/) with the indicated gene name and the above mentioned “number” was extracted from the “Most Frequent Somatic Mutations” section. This method is now described in the manuscript as legend to table 1. Please see also in Line65 in the revised manuscript.

  1. Line 55 and 60, the authors used “etc”, but I am unsure of the intended meaning. The authors should add the rest of the possibilities.

Responses:

We apology for confusing the reviewer by using “etc” in an improper way. We rewrote the sentence as “after translation, such as phosphorylation, ubiquitination, methylation, glycosylation, acetylation, nitrosylation and so on [12-14].” See also in Line70-71 in the revised manuscript. For the one in Line 60, We rewrote the sentence as “including DNA damage recognition, DNA binding ability, nuclease activity and signal transmission ability”, which emphasize only the mainly biological process regulated by PTM of MRE11. See also this in Line76 in the revised manuscript.

  1. Line 82, a reference (PMID: 28059759) can be added.

Responses:

Mimitou and colleagues did an excellent work to uncover the feature of the meiotic double-strand break resection landscape. We have included this reference (as Reference 25). See this in Line98 in the revised manuscript. Many thanks for the suggestion. 

  1. Line 134, “MRE11 and MRN play roles to repair…,” MRE11 is included in MRN complex. This should be spelled out.

Responses:

Following the reviewer’s suggestion, we removed “MRE11 and” from the sentence. this in Line165 in the revised manuscript.

  1. In the section “Biological functions of MRE11” why didn’t the authors mention VDJ and CSR? The authors should add overviews of them.

Responses:

Many thanks to the reviewer for her/his very constructive suggestion. We have now included an independent paragraph as 2.3 for an overviews of MRE11 in V(D)J and CSR. Please see this paragraph in line 130~143 in the revised manuscript.

  1. Line 414, I am unsure that D-NHEJ is commonly used in this field. D-NHEJ generally means DNA-PK (DNA-dependent protein kinase)-dependent pathway of NHEJ; although this term is not widely used in our field. To avoid any confusion for readers, the authors should simply use “deleterious NHEJ” instead of D-NHEJ throughout the manuscript.

Responses:

We completely agree with this reviewer that D-NHEJ is commonly used for DNA-PK -dependent pathway of NHEJ. Since “deleterious NHEJ” refers to alternative NHEJ (alt- NHEJ) . We therefore use “alt- NHEJ” in stead of “D-NHEJ “. Please see line 488~490 in the revised manuscript and also response to reviewer 2.’

  1. Line 417, I cannot find the origin of the abbreviation “RK/RK.”

Responses:

We apology for missing the define of RK/RK. It refers to a homozygous arginine (R) to lysine (K) mutations (RK/RK). We rewrote the first sentence of the paragraph as follow to define the abbreviation of RK/RK: “Mice, which harbor a homozygous arginine (R) to lysine (K) mutations in the GAR motif of MRE11--namely Mre11 RK/RK, …”. Please see also line 491~492 in the revised manuscript.

  1. Some editorial errors should be corrected, for example, [38][39] -> [38, 39] and tumor genesis -> tumorigenesis.

Responses:
We corrected all the editorial, such as [38][39] to [38, 39] , and spelling errors,  such as “tumor genesis” to “tumorigenesis”.

  1. The text is well written throughout the manuscript; however unfortunately, the quality of the figures is poor. Each figure does not appropriately represent the summary of the text. If space is not limited, Figure 2 should be separated to clarify the role of each PTM against MRE11, i.e., structure (Figure 2), phosphorylation (Figure 3), ubiquitination (Figure 4), and methylation (Figure 5). For example, by looking at Figure 2 in the current version, it is unclear whether the phosphorylation of MRE11 activates its nuclease activity or the recruitment of MRN at DSB sites. And also, for example, it is unclear whether cell viability and tumorigenesis are downstream of HR and NHEJ. I am unsure why these are drawn in parallel arrows. In addition, if a PTM affects other PTM events, the interplay should be also discussed in the text or figures.

Responses:
We appreciate the constructive comments from this reviewer regarding to the Fig 2. Indeed, some biological function of MRE11 haven’t bee presented and the stream of biological consequence caused by PTM of MRE11 is not clear. We take the suggestion from this review and split Fig2 into Fig3~6. See Line 244, Line 289, Line 384 and Line 477 in the revised manuscript.

It is a very interesting topic about the interplay among different PTM to MRE11. Unfortunately, there is nearly no experimental evidences for the interplay among different PTMs. Therefore, we discuss this point in Line 548. 

Reviewer 2 Report

In this manuscript Ruiqing Lu and coll. review the post-translational modifications (PTMs) of Mre11. First, they describe the biological functions of Mre11, the relation between Mre11 and human diseases, and then in a third part, the PTMs of Mre11 and their biological and/or pathological implications. In this paragraph, they describe the phosphorylations, ubiquitinations and  ubiquitin-like modifications, and finally methylations of Mre11.

Globally this review is well written, and well organized. It will certainly be useful for people who are looking for information about the PTMs of Mre11 and their implications.

The only criticism I have would be that a paragraph dealing with the reversibility of these PTMs would be useful. Indeed, the authors argue that these modifications are rapid and are therefore efficient switches to regulate important cellular processes like DNA damage repair or DNA damage response, which I agree with. But in that respect, switches need to be turned off and a paragraph on how these modifications are reversed -or at least what is known about that up today- would be useful.

Comments:

General comment: at no point the partner CtIP is mentioned when it is a crucial factor in the initiation of resection, the removal of protein-DNA adducts, etc...

Table 1: would that be feasible to indicate what part of these mutations are detected in sporadic cancers and what part in hereditary cancers?

Line 77 : I don’t understand what the authors mean by Spo11 creating an “incomplete DSB”? Spo11 is a topo2-like enzyme that cuts the DNA to create a DSB. So it is a real, complete DSB?

Figure 1 : Panel A: the figure is relatively loaded and the correspondence between the central circle and the illustrations around it are not always obvious : for example there is a mix between the response to viral infection and the stabilization of telomeres, or between the cohesion of sister chromatid (or what is supposed to be “”sister chromatid” and not sister chromosome” as indicated on the figure) and the stabilization of replication forks.

It would be good to cite the papers where these functions of Mre11 have been described (for example in the legend of the figure).

Panel B: Could the authors add on the figure the interaction domains with Nbs1 and RAD50? The explanation of RBD is missing in the legend.

Table 2: please cite the papers.

Paragraph 3: please discuss cancer predisposition in all the mentioned syndromes. Are mutations of Mre11 implicated in hereditary cancers?

Line 205s : ‘phosphorylation of Mre11 independent of ATM, likely via other kinases under certain circumstances’ sounds not only obvious (if it’s not ATM it is another kinase and it is necessarily in “some”circumstances”) but also extremely vague. Could you please describe more precisely what kinases and in what context?

Line 239: I don’t agree with the statement that ‘it is widely accepted that HR and NHEJ pathways compete’: actually NHEJ competes with the initiation of resection of DNA ends (which is actually mediated by Mre11/ CtIP). This resection can lead to HR of course but also to SSA, A-EJ, MM-BIR, TMEJ, MMEJ, etc…

Line 242 : the sentence about exo1 is too long and too complicated : sorry, but at the end I don’t understand if phosphorylation of Mre11 is necessary for resection mediated by Exo1 or the opposite. Could you please make it simpler?

Line 257: Just a comment: the regulation of Mre11 by cell cycle regulation kinases can indeed suggest a role of Mre11 phosphorylation in cell cycle but also a role of cell cycle in the regulation of Mre11 activity.

Line 414: I have never heard about this term ‘Deletional-NHEJ’. Although  it sounds right, I recommend to use a widely admitted terminology and choose among the classic terms A-EJ, alt-NHEJ, etc…

Typos and writing issues:

Line 35: the sentence “HR is more complex with highly accurate and error free” seems incomplete.

Line 179 : typo on “However”

Line 196 : the end of the sentence is missing.

Line 282: ‘MRE11 is phosphorylated S676’

Line 284: ‘leads to accumulated DSB and to fuel ‘

Line 373: ‘SUMOylation regulates’

Line 379: ‘reversible ways’

Line 484: ‘MRE11-associated’

Author Response

Reply to Review#2
Comments and Suggestions for Authors

In this manuscript Ruiqing Lu and coll. review the post-translational modifications (PTMs) of Mre11. First, they describe the biological functions of Mre11, the relation between Mre11 and human diseases, and then in a third part, the PTMs of Mre11 and their biological and/or pathological implications. In this paragraph, they describe the phosphorylations, ubiquitinations and ubiquitin-like modifications, and finally methylations of Mre11.

Globally this review is well written, and well organized. It will certainly be useful for people who are looking for information about the PTMs of Mre11 and their implications.

Responses:

We thank the reviewer for her/his positive evaluation of our manuscript.

  1. The only criticism I have would be that a paragraph dealing with the reversibility of these PTMs would be useful. Indeed, the authors argue that these modifications are rapid and are therefore efficient switches to regulate important cellular processes like DNA damage repair or DNA damage response, which I agree with. But in that respect, switches need to be turned off and a paragraph on how these modifications are reversed -or at least what is known about that up today- would be useful.

Responses:

Many thanks again to the reviewer for her/his very constructive suggestion. Indeed, we believe that the balance of PTMs by adding and removing is very important and interesting. Although it has been shown that some reversibility of Mre11 PTMs happen, such as dephosphorylation, but the biological functions of the reversible -process are still unclear. Therefore, we include this point as a direction of further prospect in Line552 in revise manuscript.

Comments:

  1. General comment: at no point the partner CtIP is mentioned when it is a crucial factor in the initiation of resection, the removal of protein-DNA adducts, etc...

Responses:

Indeed, CtIP is very important factor to imitate the resection of DSBs. We did cite the article deal with CtIP without mentioned it in the text. We have now included it (such as line138, line166) in the revised manuscript.

  1. Table 1: would that be feasible to indicate what part of these mutations are detected in sporadic cancers and what part in hereditary cancers?

Responses:

According to the information from TGCA database, these mutations are all somatic mutations. We now emphasize this point in the table. See line 56  in in the revised manuscript.

  1. Line 77 : I don’t understand what the authors mean by Spo11 creating an “incomplete DSB”? Spo11 is a topo2-like enzyme that cuts the DNA to create a DSB. So it is a real, complete DSB?

Responses:

Meiotic recombination initiates with DSB covalent with Spo11, which attacks DSB 5′ ends and forms protein-DNA adduct. “incomplete DSB” is a misleading word for the protein-DNA adduct at DSB site. We therefore changed “incomplete DSB” to “protein-DNA adduct” .See this in line 94 in the revised manuscript.

  1. Figure 1 : Panel A: the figure is relatively loaded and the correspondence between the central circle and the illustrations around it are not always obvious : for example there is a mix between the response to viral infection and the stabilization of telomeres, or between the cohesion of sister chromatid (or what is supposed to be “”sister chromatid” and not sister chromosome” as indicated on the figure) and the stabilization of replication forks.

Responses:
Following the suggestion of this reviewer, we have split the around illustrations together with correspondence central circle contain as fig 1a…g, removed the confusing link between viral infection and telomere stabilization, as well as corrected the typo mistakes.

  1. It would be good to cite the papers where these functions of Mre11 have been described (for example in the legend of the figure).

Responses:
We have split the around illustrations together with correspondence central circle contain as fig 1A-G and also cite figure separately in each paragraph of text.

  1. Panel B: Could the authors add on the figure the interaction domains with Nbs1 and RAD50? The explanation of RBD is missing in the legend.

Responses:
Following the suggestion from this review, we added interaction domains with NBS1 and RAD50, and also explained the “RBD” on the figure (as fig 2A). RBD is interaction domain between RAD50 and MRE11.

  1. Table 2: please cite the papers.

Responses:
These data were first collected from database cBioPortal and summary in table 2. We now included the method for collecting data was described in the revised manuscript in table legend. We also cite the papers (PMID: 23550210) which developed cBioPortal.

  1. Paragraph 3: please discuss cancer predisposition in all the mentioned syndromes. Are mutations of Mre11 implicated in hereditary cancers?

Responses:
Paragraph 3 focuses mainly more on PTMs of MRE11. We added the discussion of cancer predisposition of mentioned syndromes in Paragraph 2 (Line48-50).  Except the ATLD syndrome, there is no Mre11 mutation in hereditary cancers was reported, to our knowledge.

  1. Line 205s : ‘phosphorylation of Mre11 independent of ATM, likely via other kinases under certain circumstances’ sounds not only obvious (if it’s not ATM it is another kinase and it is necessarily in “some”circumstances”) but also extremely vague. Could you please describe more precisely what kinases and in what context?

Responses:
We apology for the confusing sentence. We rewrote the sentence as: “likely via other kinases, such as ATR, upon DSBs produced by HaeIII digestion”. See line 265 in the revised manuscript.

  1. Line 239: I don’t agree with the statement that ‘it is widely accepted that HR and NHEJ pathways compete’: actually NHEJ competes with the initiation of resection of DNA ends (which is actually mediated by Mre11/ CtIP). This resection can lead to HR of course but also to SSA, A-EJ, MM-BIR, TMEJ, MMEJ, etc…

Responses:
We rewrote the sentence to : “it is widely accepted that HR and NHEJ are two mainly pathways for the repair of DSBs”. See line 294 in revised manuscript.

  1. Line 242 : the sentence about exo1 is too long and too complicated : sorry, but at the end I don’t understand if phosphorylation of Mre11 is necessary for resection mediated by Exo1 or the opposite. Could you please make it simpler?

Responses:
The phosphorylation of Mre11 plays role to limit ATRM-dependent phosphorylation of EXO1. Defect of Mre11 phosphorylation could accelerate the exonuclease of EXO1 to resect DNA therefore block HR. We rewrote this sentence in the revised manuscript as: “It is worth noting that the phosphorylation of Mre11 plays role to limit ATM-mediated phosphorylation of EXO1. Impair of EXO1 phosphorylation can accelerate exonuclease activity of EXO1 and lead to uncontrolled resection, which can block HR repair.” Please see line 297~301 in the revised manuscript.

  1. Line 257: Just a comment: the regulation of Mre11 by cell cycle regulation kinases can indeed suggest a role of Mre11 phosphorylation in cell cycle but also a role of cell cycle in the regulation of Mre11 activity.

Responses:
We completely agree with this reviewer. We therefore rewrote the sentence as “suggesting that phosphorylation of Mre11 may play role in cell cycle regulation”. Please see line 321~313 in the revised manuscript.

  1. Line 414: I have never heard about this term ‘Deletional-NHEJ’. Although  it sounds right, I recommend to use a widely admitted terminology and choose among the classic terms A-EJ, alt-NHEJ, etc…

 Responses:

We completely agree with this reviewer and now we use “alt- NHEJ” in stead of “D-NHEJ “throughout the revised manuscript. Please see line 488~490 in the revised manuscript and also response to reviewer 1 Point 7.

  1. Typos and writing issues:

Line 35: the sentence “HR is more complex with highly accurate and error free” seems incomplete.

 Responses:
We have re-organized the paragraph and changed the sentence to “ HR is a complex and relatively slow process involving multi-steps, which repairs DSBs mainly occurring in S-G2 phases with high-fidelity [7]” Please see line 30~35 in the revised manuscript and also response to reviewer 1 Point 1.

Line 179 : typo on “However”

 Responses:
Has be corrected!

Line 196 : he end of the sentence is missing.

 Responses:
The rest of the sentence “14 sites have been clarified biological functions.” was added. lease see line 242~243 in the revised manuscript.

Line 282: ‘MRE11 is phosphorylated S676’

 Responses:
Has been corrected to “MRE11 is phosphorylated at S676”.

Line 284: ‘leads to accumulated DSB and to fuel ‘

 Responses:
Corrected the phrase “to fuel” to “fueling”.

Line 373: ‘SUMOylation regulates’

 Responses:
“regualtes” has been corrected to “regulates”. See line439 in in the revised manuscript.

Line 379: ‘reversible ways’

 Responses:
Corrected!

Line 484: ‘MRE11-associated’

  Responses:
“assocaited” has been corrected “to “associated”.